# RETHINKING THE VALUE OF PROMPT LEARNING FOR VISION-LANGUAGE MODELS

## ABSTRACT

Large-scale visual-language pre-training like CLIP has demonstrated great success in open-set visual concept learning that enables zero-shot transfer to downstream tasks through prompting. To automate prompt engineering, prompt learning is proposed to automatically learn the optimal task-relevant prompts. In this paper, we make some surprising observations that contradict common beliefs about prompts. We observe that even random prompts can achieve pretty good performance for zero-shot recognition. We also find that prompt learning gives comparable or worse performance than directly fine-tuning of the linear classifier. Moreover, prompt learning is no more than parameter-efficient learning, and is a trade-off between optimality and generalization. Our results highlight the need for the rethinking of existing prompt learning, more careful baseline evaluations in future research on prompt learning methods in vision-language models.

## 1 INTRODUCTION

Building a state-of-the-art visual recognition system is one of the core tasks in the field of computer vision. Current state-of-the-art visual recognition systems are almost all based on Deep Neural Networks (DNNs), which can be roughly divided into two parts: a non-linear feature extractor and a linear classifier. For traditional visual recognition, where the class number are fixed and the labels are discretized, the standard practice is to assign each category with a weight vector, which is optimized to maximize the classification accuracy. Take the ResNet for ImageNet classification as an example, the weight vectors for 1000 classes form the weight matrix $W \in \mathbb{R}^{1000 \times 4096}$ of the linear classifier (the last fully-connected layer of ResNet), where 4096 is the dimension of the features from the feature extractor. This learning paradigm can only learn closed-set visual concepts related to the pre-defined categories, and can not generalize to new classes beyond these closed-set categories.

In contrast to supervised learning with fixed labels of a closed-set categories, visual concept learning with the supervision of text has shown great potential. The main inspiration is that language is a high level abstraction of human understanding the world, thus it contains rich information and can naturally generalize well. One of the representative works is the CLIP (Contrastive Language-Image Pretraining) (Radford et al., 2021), which learns joint representations of vision and language using contrastive learning on large-scale image and text data. Thanks to the rich information and the generality of natural language, the CLIP model can learn diverse and task-agnostic visual-textual representations, which can be generalized to many downstream tasks even under the zero-shot setting. This is done by using the names of all classes of a downstream task as the text for textual feature extraction, and conducting classification based on the alignment score of the visual features and the textual features for each class. However, using the class names as the text is deficient due to the lack of context. To this end, the authors of Radford et al. (2021) resort to the technique of *prompt tuning* (Liu et al., 2021a). Here the "prompt" is a cloze templates which specifies the context about the task at hand. They find that the template "`a photo of a {CLASS}.`" is a good prompt for image classification. By using elaborate prompt engineering and ensemble, much higher zero-shot performance can be achieved.

Prompt engineering has shown greater transferability than the contextless baseline of using class names. The drawback is that the handcrafted prompt tuning requires prior knowledge about the downstream task. Moreover, as pointed out in Zhou et al. (2022b), the performance is very sensitive

to a slight change in the wording of the prompt template. Thus prompt tuning is a non-trivial task. To solve this problem, the authors of Zhou et al. (2022b) bring the concept of prompt learning from natural language processing (NLP) and propose Context Optimization (CoOp) to automate the prompt engineering in vision-language models. More recent works including (Ju et al., 2021; Yao et al., 2021; Zhou et al., 2022a) are continually developed. The core idea of these prompt learning approaches is to treat the embeddings of the words in a prompt as a set of learnable vectors, which are learned through back-propagation w.r.t. the downstream task loss.

Prompts can encode context information expressed in natural language about the target tasks, thus they can generalize well and show promising results even in zero-shot. Prompt learning, which automatically optimize the prompts in the same word embedding space of natural language, is believed to have two advantages. First, it is believed that prompt learning converge faster and requires fewer training examples than fine-tuning. This is because only the context vectors are updated while the pre-trained parameters of both text encoder and image encoder are fixed. Moreover, during the gradients calculation, the pre-trained knowledge encoded in the text encoder can also be back-propagated through the network to the context vectors. Therefore, prompt learning is commonly believed to be superior to linear probe, partial fine-tuning or even full fine-tuning. Second, it is believed that the learned prompts have strong robustness and generalization ability, as the optimization is conducted in the NLP embedding space, thus the learned prompts are expected to provide high generalization ability in the same way as natural language.

In this paper, we test these two beliefs by evaluating the prompt tuning/learning performance of CLIP on various downstream tasks. We start from examining the influence of text encoder on the prompts through handcrafted prompts and random prompts and show that the text encoder can indeed provide some regularization on the prompts. To our surprise, we find that even random prompts can still achieve pretty good performance for zero-shot recognition. Then, we compare prompt learning and fine-tuning for closed-set recognition, and observe that prompt learning gives comparable or worse performance than directly fine-tuning the weights of the linear classifier. Last, we examine the generalization ability of the learned prompts, and reveal that prompt learning is no more than parameter-efficient learning, and is a trade-off between optimality and generalization.

## 2 RELATED WORKS

Prompt learning is originally proposed to transfer knowledge from pre-trained language models to downstream tasks, which has demonstrated great performance in NLP domain Devlin et al. (2018); Brown et al. (2020). A typical example of prompt learning is "fillin-the-blank" cloze templates Petroni et al. (2019), which transforms the down-stream task to a format familiar to the pre-trained model. Instead of manually designing prompt templates, later studies focus on automated prompt learning which can be categorized into discrete prompts and continuous prompts Liu et al. (2021a). Researchers discover the discrete prompts in a discrete space, e.g. natural language phrases, and most works generate discrete prompts by either gradient-based search Wallace et al. (2019), or prompt mining Jiang et al. (2020), or prompt generation Gao et al. (2020), etc. Instead of limiting the prompt to human-interpretable natural language domain, continuous prompts in the embedding space of the model are proposed. Several representative methods on continuous prompts learning include prefix tuning Li & Liang (2021), tuning initialized with discrete prompts Zhong et al. (2021), and hard-soft prompt hybrid tuning Liu et al. (2021b).

Motivated by the well performance of prompt learning on NLP, recently researchers begin to apply it into the vision-language models. CLIP Radford et al. (2021) uses a manually designed prompt on the text encoder, which enables the zero-shot image classification of vision-language model. To avoid human efforts on prompt design, CoOp Zhou et al. (2022b) proposes a continuous prompts learning method and two implementations that can be applied on different recognition tasks. Yet CoOp Zhou et al. (2022b) seems over-fitting the base classes in the training, resulting in inferior performance on unseen classes even within the same dataset. To cure this problem, CoCoOp Zhou et al. (2022a) propose to generate an input-conditional vector for each image by a lightweight neural network, which boosts the classifier performance on new classes. Although CoOp and CoCoOp achieve promising improvements, they requires supervised data from the target datasets which may restrict the model scalability. In the contrary, Huang et al. Huang et al. (2022) propose the unsupervised prompt learning (UPL) method which improves transfer performance of CLIP-like VL models

without labeled data. Different from above prompt learning methods which apply the prompts on the text encoder of VL model, VPT Jia et al. (2022) uses prompts learning on the image encoder. Specifically, they prepend a small amount (less than 1% of model parameters) of trainable parameters into the input sequence of transformer layers, and keep the model backbone frozen. Besides the prompt learning for the classification task, there are some studies about transferring knowledge from VL models to other downstream tasks, such video understanding Ju et al. (2021), object detection Du et al. (2022), and visual grounding Yao et al. (2021).

## 3 PROMPT LEARNING BASED ON CLIP

The analysis throughout this paper is based on CLIP model (Radford et al., 2021), which consists of an image encoder $f(\cdot)$ and a text encoder $g(\cdot)$. The image encoder is usually a ResNet or a ViT, while the text encoder is a Transformer. Through contrastive learning, the two encoders are trained to transform input images and texts into the same feature space.

The aligned visual-textual feature space makes CLIP to be capable of zero-shot image recognition. Specifically, the input image $x$ is feed into the image encoder to obtain the visual representation $f$. Similarly, for each category, the class name (e.g., "dog") wrapped in the prompt template (e.g., "a photo of a {CLASS}.") is feed into the text encoder to obtain the textual representation $\{w_i\}_{i=1}^K$, where $K$ is the class number. Then the prediction probability is as follows:

$$p(y = i|x) = \frac{\exp(\cos(f, w_i)/T)}{\sum_{j=1}^K \exp(\cos(w_j, w_j x)/T)}, \tag{1}$$

where $T$ is the Softmax temperature and $\cos(\cdot, \cdot)$ denotes cosine similarity.

To ease the prompt engineering process, the concept of prompt learning is proposed. Context Optimization (CoOp) (Zhou et al., 2022b) is one of the earliest works that introduce prompt learning to adapt pre-trained vision-language models to downstream tasks. The key idea of prompt learning is to automatically learn the prompt template instead of using a handcrafted template. Specifically, CoCop introduces a set of learnable prompt vectors with the following format,

$$t = [v]_1[v]_2 \cdots [v]_M[v]_{\text{CLASS}}, \tag{2}$$

where $\{[v]_m, m = 1, \cdots, M\}$ are the set of the learnable word embeddings of the prompt template, which are shared for all classes, $M$ is the number of context tokens of the prompt, and $[v]_{\text{CLASS}}$ is the embedding of the class name. For each class $i \in 1, \cdots, K$, we can obtain the prompt $t_i$ according to Eq. (2). Then the prediction probability is as follows:

$$p(y = i|x) = \frac{\exp(\cos(f, g(t_i))/T)}{\sum_{j=1}^K \exp(\cos(f, g(t_j))/T)}, \tag{3}$$

With the prediction probability of Eq. (3) and the classification loss of the downstream task, we can optimize the learnable prompt vectors $\{[v]_m, m = 1, \cdots, M\}$ while frozening the pre-trained weights of the CLIP. Specifically, the gradients w.r.t. the learnable prompt vectors can be back-propagated all the way through the text encoder $g(\cdot)$, at which time, the pre-trained knowledge encoded in the text encoder of CLIP can be distilled to the learnable prompt vectors. In this way, the learned prompts can encode some useful information about the downstream task.

## 4 TEXT ENCODER AS A REGULARIZATION

In section 3, we show that different prompts would result in different weights for classification, i.e., $w_i = g(t_i)$, for $i = 1, \cdots, K$. In this section, we examine how different prompts would affect the performance of classification across a wide range of datasets.

### 4.1 HANDCRAFTED PROMPTS

We start from handcrafted prompts. Previous works have shown that a good handcrafted prompt could greatly improve zero-shot classification accuracy Radford et al. (2021). In this section, we evaluate the performance of zero-shot classification with various handcrafted prompt templates.

Table 1: Various handcrafted prompts.

| Prompt Type | Prompt Template |
|---|---|
| ClassName | "{CLASS}" |
| Basic | "a photo of a {CLASS}." |
| Revised | "this is a photo of a {CLASS}." |
| Negative | "this is not a photo of a {CLASS}." |

Table 2: Results of zero-shot classification with various handcrafted prompts and random prompts across the 11 datasets.

| Dataset | ClassName | Basic | Revised | Negative | RandToken | RandEmbed |
|---|---|---|---|---|---|---|
| ImageNet | 55.3 | 58.2 | **59.2** | 58.0 | $50.0_{\pm2.7}$ | $47.0_{\pm2.8}$ |
| Caltech101 | 80.9 | 85.9 | **87.0** | 84.6 | $75.3_{\pm3.4}$ | $74.7_{\pm5.4}$ |
| OxfordPets | 78.8 | 83.7 | **84.7** | 81.0 | $73.2_{\pm3.8}$ | $72.3_{\pm3.4}$ |
| StanfordCars | 54.4 | 55.6 | **55.6** | 49.0 | $52.6_{\pm0.9}$ | $46.5_{\pm2.5}$ |
| Flowers102 | 57.3 | 60.9 | **62.8** | 62.7 | $51.4_{\pm2.6}$ | $46.7_{\pm7.6}$ |
| Food101 | 73.9 | 75.3 | **77.2** | 75.5 | $72.8_{\pm3.1}$ | $69.0_{\pm2.8}$ |
| FGVCAircraft | 15.3 | 15.7 | **15.8** | 15.0 | $12.2_{\pm2.5}$ | $10.2_{\pm2.1}$ |
| SUN397 | 54.9 | **58.5** | 57.7 | 58.5 | $47.6_{\pm3.3}$ | $45.0_{\pm2.4}$ |
| DTD | 41.1 | 40.0 | **41.1** | 42.3 | $34.1_{\pm5.6}$ | $28.6_{\pm3.4}$ |
| EuroSAT | **28.4** | 24.2 | 28.0 | 36.7 | $22.4_{\pm3.9}$ | $24.2_{\pm4.5}$ |
| UCF101 | 56.4 | **58.3** | 57.9 | 57.7 | $52.5_{\pm2.4}$ | $47.9_{\pm2.2}$ |

First, we want to evaluate the prompts used by current prompt-based methods. The simplest baseline is to directly use the class name as the input of the text encoder (ClassName). To improve performance, the authors of CLIP Radford et al. (2021) propose a basic prompt template for image recognition (Basic). The authors of Ju et al. (2021) further revise the prompts by adding a "this is" prefix (Revised). Second, we want to examine what will happen if we use the "negative prompt" for the classification task, i.e., by adding a "not" in the prompt template (Negative). For clearness, we summarize the handcrafted templates studied in this section in Table 1.

Following (Zhou et al., 2022b), we conduct experiments on the 11 image classification datasets used in CLIP which are publicly available, i.e., ImageNet (Deng et al., 2009), Caltech101 (Fei-Fei, 2004), OxfordPets (Parkhi et al., 2012), StanfordCars (Krause et al., 2013), Flowers102 (Nilsback & Zisserman, 2008), Food101 (Bossard et al., 2014), FGVCAircraft (Maji et al., 2013), SUN397 (Xiao et al., 2010), DTD (Cimpoi et al., 2014), EuroSAT (Helber et al., 2019) and UCF101 (Soomro et al., 2012). The results are summarized in Table 2.

As expected, the results in Table 2 show that the ClassName prompt template achieves the worst performance. On only one dataset (EuroSAT), the ClassName could outperforms the Basic and the Revised prompt. There is a large accuracy improvement using the Basic prompt template introduced in Radford et al. (2021). The accuracy can be further improved when using the Revised template (Ju et al., 2021), which achieves the best accuracy for 8 out of the 11 datasets.

An important finding we want to point out is that the relative performance for these three handcrafted prompts are somewhat consistent across various dataset. Specifically, the Basic template outperforms the ClassName prompt for 9 out of the 11 datasets, and the Revised template outperforms the Basic template for other 9 out of the 11 datasets. This finding indicates that the handcrafted prompts expressed in natural language can generalize well across various datasets. In other word, a prompt template works on one dataset has a high probability to also work on another dataset.

Another surprising observation is that the negative prompt can also achieve very good performance. On most of the datasets, the Negative prompt shows quit large improvement over the ClassName template. By comparing with the other handcrafted prompts, it even get the best accuracy on 3 out of the 11 datasets.

## 4.2 RANDOM PROMPTS

Then we evaluate what would happen if random prompts are used. Here we consider two kinds of random prompts, namely the random token template and the random embedding template:

- **Random Token Template** prepends some random word IDs which are selected from the 49,152 vocabulary of CLIP.
- **Random Embedding Template** prepends some random embedding vectors after transforming token IDs into word embedding.

Similarly, we evaluate the performance across 11 image classification datasets. Note that the selection of the random seed would make a large difference in performance. So we run the experiments for 10 times with different random seeds and report the average performance. The results are shown in the last columns of Table 2. To our surprise, the results show that even random prompts can still achieve pretty good performance for zero-shot recognition. It seems that the text encoder can indeed provide some regularization at the encoding process of the input text with prompts.

## 4.3 SUMMARY

By now, we have evaluated the performance of different handcrafted prompts and random prompts. In the following list, we summarize our findings, some of which are quit surprising.

1. Handcrafted prompts expressed in natural language show great power and generalization ability, and the relative performance for different prompt templates are somewhat consistent across a wide range of datasets.
2. The negative prompts, which provide wrong context information for the downstream tasks, can also achieve very good performance.
3. Even random prompts can still achieve pretty good performance for zero-shot recognition.

## 5 PROMPT LEARNING V.S. CLASSIFIER FINE-TUNING

In this section, we evaluate the effect of prompt learning for closed-set image classification task, where the training set and testing set are from the same categories. For closed-set classification, it is widely believed that prompt learning converge faster and requires fewer training examples than fine-tuning. The reasons are two-fold. First, in prompt learning, only the prompt vectors are learned while the pre-trained CLIP model is fixed. Second, as shown in Eq. 3, to optimize the prompt vectors $\{[\boldsymbol{v}]_m, m = 1, \cdots, M\}$, the gradients need to be back-propagated all the way through the text encoder $g(\cdot)$. This process allows the knowledge learned by the CLIP model to be distilled from the weights to the prompts (Zhou et al., 2022b).

In this paper, we challenge this common belief. By comparing Eq. 1 and Eq. 3, it is easy to see that $\boldsymbol{w}_i = g(\boldsymbol{t}_i)$ for $i = 1, \cdots, K$, which can be viewed as the weights of the last classifier. At inference time, we first need to feed each $\boldsymbol{t}_i$ into the text encoder to obtain $\boldsymbol{w}_i$, then the generated weights $\{\boldsymbol{w}_i, i = 1, \cdots, K\}$ are used for classification. Our key question is that *if we could directly optimize $\boldsymbol{w}_i$, why should we optimize the latent vector $\boldsymbol{t}_i$?* To this end, we conduct comprehensive experiments on the 11 image classification datasets to see if prompt learning (optimize $\boldsymbol{t}_i$) is superior to classifier fine-tuning (optimize $\boldsymbol{w}_i$).

## 5.1 TRAINING DETAILS

For prompt learning, we utilize the Context Optimization (CoOp) method proposed in Zhou et al. (2022b), which is one of the earliest works that introduce prompt learning to adapt pre-trained vision-language models to downstream tasks. We use the CLIP pre-trained model with ResNet-50 as the image encoder. The number of learnable prompt vectors $M$ is set to 16 and are shared across call categories, which is the default setting for in the original paper of Zhou et al. (2022b). During the experiments, we find that the results for prompt learning on downstream tasks are very sensitive to the choice of hyper-parameters. For fair comparison, we use the grid search over the training epochs and learning-rate and report the best accuracy for all experiments. Specifically, the number

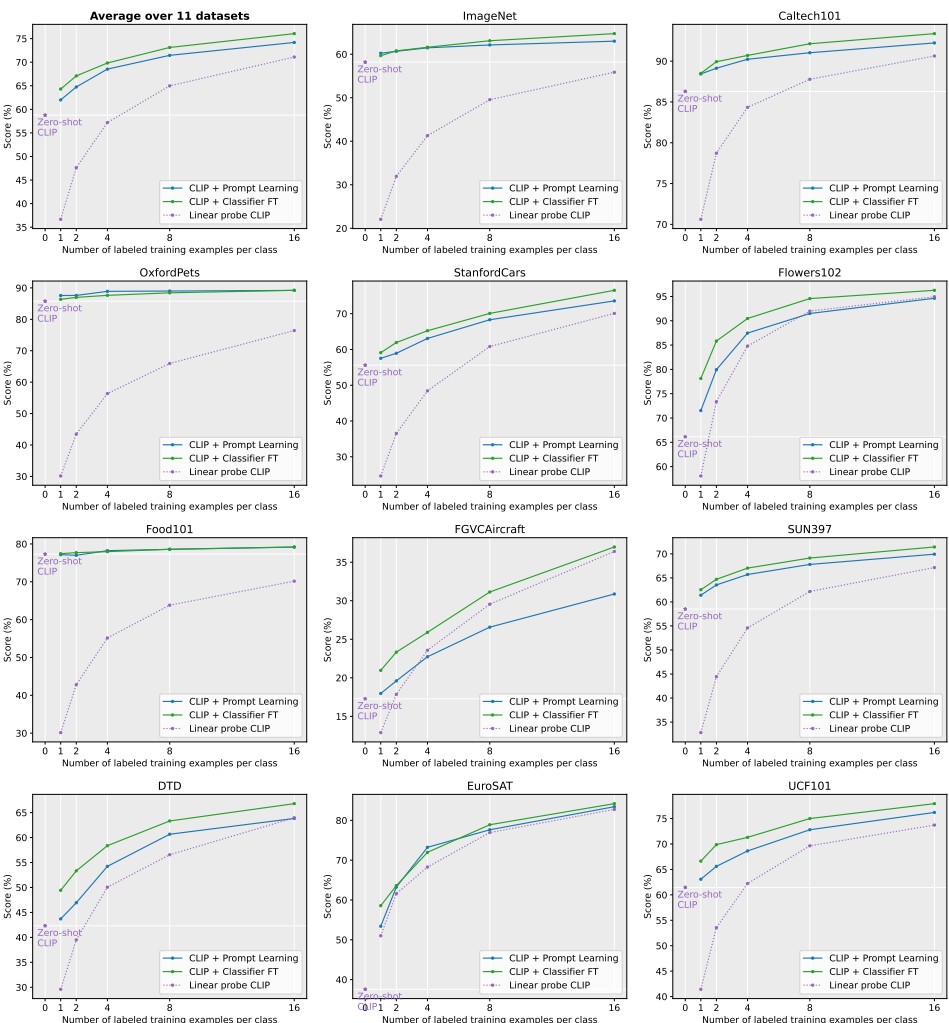

Figure 1: Comparison between prompt learning (CLIP + Prompt Learning) and classifier fine-tuning (CLIP + Classifier FT) of few-shot learning on the 11 datasets. The results for zero-shot CLIP (stars) and linear probe (dashed lines) are also given.

of training epoch is set to $\{50, 100, 200\}$. The starting learning-rate is set to $\{2e-2, 2e-3, 2e-4\}$ and $\{2e-1, 2e-2, 2e-3\}$ for prompt learning and classifier fine-tuning, which are the best choices for both cases. Other hyper-parameters are the same as in Zhou et al. (2022b).

## 5.2 RESULTS

We compare the performance of prompt learning and classifier fine-tuning across all 11 datasets. Following Zhou et al. (2022b), we report few-shot learning results with 1/2/4/8/16 shots for training while using the original test set for testing. All the results for prompt learning and classifier fine-tuning are the average over 3 runs with different random seeds. We also give the zero-shot results using CLIP model as well as the linear probe results based on CLIP for comparison. The main results are shown in Figure 1.

From Figure 1, we can see that both the prompt learning and classifier fine-tuning can dramatically outperforms the zero-shot and the linear probe based on CLIP for most of experiments. Compared with prompt learning, the simple classifier fine-tuning cal obtain much higher accuracy, except for the OxfordPets dataset, on which the classifier fine-tuning is slightly inferior to prompt learning. The average results on 11 datasets displayed in the top-left corner of Figure 1 show that classifier

Table 3: The optimization time (in minutes) of prompt learning and classifier fine-tuning with various backbones for 16 shots ImageNet classification.

| Method | ResNet-50 | ResNet-101 | ViT-B/32 | ViT-B/16 |
|---|---|---|---|---|
| Prompt Learning | 139.5 | 143.3 | 136.3 | 143.1 |
| Classifier Finetuning | 13.8 | 15.9 | 13.7 | 14.8 |

Table 4: Comparison between prompt learning and classifier fine-tuning on robustness to distribution shift using different vision backbones. Bold value indicates the best result.

| Method | Source | Target | | | | Average |
|---|---|---|---|---|---|---|
| | ImageNet | -V2 | -Sketch | -A | -R | |
| **ResNet-50** | | | | | | |
| Zero-Shot CLIP | 58.18 | 51.34 | 33.32 | 21.65 | 56.00 | 44.10 |
| Linear Probe CLIP | 55.87 | 45.97 | 19.07 | 12.74 | 34.86 | 33.70 |
| Prompt Learning | 63.00 | 55.27 | 34.03 | **22.40** | 55.90 | 46.12 |
| Classifier Fine-tuning | **64.73** | **56.03** | **34.13** | 22.10 | **58.30** | **47.06** |
| **ResNet-101** | | | | | | |
| Zero-Shot CLIP | 61.62 | 54.81 | 38.71 | 28.05 | **64.38** | 49.51 |
| Linear Probe CLIP | 59.75 | 50.05 | 26.80 | 19.44 | 47.19 | 40.65 |
| Prompt Learning | 66.53 | **58.73** | 40.00 | **29.00** | 64.03 | 51.66 |
| Classifier Fine-tuning | **67.50** | 58.60 | **40.37** | 28.90 | 64.20 | **51.91** |
| **ViT-B/32** | | | | | | |
| Zero-Shot CLIP | 62.05 | 54.79 | 40.82 | 29.57 | 65.99 | 50.64 |
| Linear Probe CLIP | 59.58 | 49.73 | 28.06 | 19.67 | 47.20 | 40.85 |
| Prompt Learning | 66.80 | **58.43** | 40.97 | **31.30** | 65.33 | 52.57 |
| Classifier Fine-tuning | **68.10** | 58.17 | **41.47** | 30.63 | **67.60** | **53.19** |
| **ViT-B/16** | | | | | | |
| Zero-Shot CLIP | 66.73 | 60.83 | 46.15 | 47.77 | 73.96 | 59.09 |
| Linear Probe CLIP | 65.85 | 56.26 | 34.77 | 35.68 | 58.43 | 50.20 |
| Prompt Learning | 71.97 | 64.40 | **47.97** | **49.97** | 75.03 | 61.86 |
| Classifier Fine-tuning | **72.97** | **64.47** | **47.97** | 48.50 | **75.70** | **61.92** |

fine-tuning is not as vulnerable as we think in the few shot setting. On the contrary, for 1 shot to 16 shots, classifier fine-tuning consistently outperforms prompt learning by 1.5% to 2.3%.

Another difference between prompt learning and classifier fine-tuning is about the optimization time. Prompt learning need to propagate the gradients through the text encoder back to the learnable prompt vectors, which is quit time-consuming. By contrast, the classifier fine-tuning can directly optimize the weights of the classifier, thus it is much more efficient. Here we report the optimization time of prompt learning and classifier fine-tuning with various network backbones for 16 shots classification on ImageNet. The results are shown in Table 3. We can observe that classifier fine-tuning is about $10\times$ more efficient in speed than prompt learning.

These results have confirmed our assumption about prompt learning for closed-set classification. Specifically, prompt learning can not achieve the goal of sample efficient training as commonly expected. The simple baseline of classifier fine-tuning is much time efficient and achieves much higher accuracy than prompt learning across various datasets.

## 5.3 ROBUSTNESS TO DISTRIBUTION SHIFTS

Previous works have shown that prompt learning has high domain generalization ability compared with handcrafted prompts. Thus we need to compare the robustness of classifier fine-tuning and prompt learning with respect to distribution shifts across domains. To this end and following Zhou

et al. (2022b), we use the ImageNet as the source domain and use the ImageNetV2 (Recht et al., 2019), ImageNet-Sketch (Wang et al., 2019), ImageNet-A (Hendrycks et al., 2021b), and ImageNet-R (Hendrycks et al., 2021a) as the target domain. These datasets have the compatible class names with ImageNet, thus the optimized prompts of prompt learning and the learned weights of classifier fine-tuning can be transfered from ImageNet to these datasets.

We summarize the results in Table 4. As we can see, despite exposure to the source dataset, both prompt learning and classifier fine-tuning outperform the zero-shot and linear probe CLIP for the target datasets, which demonstrates their strong robustness to distribution shift. Moreover, classifier fine-tuning surpasses prompt learning on most models and datasets, verifying the generalization advantage of classifier fine-tuning over prompt learning.

# 6 OPTIMALITY-GENERALIZATION TRADE-OFF

In this section, we examine the generalization ability of prompt learning method. It is usually believed that the learned prompts have strong robustness and generalization ability, as the optimization is conducted in the NLP embedding space, thus the learned prompts are expected to provide high generalization ability in the same way as natural language. However, as pointed out in Zhou et al. (2022a), the prompt learning method used in Zhou et al. (2022b) fails to learn task-specific context that generalizes well to unseen classes. To solve this problem, conditional prompt learning (Conditional Context Optimization, CoCoOp) is proposed in Zhou et al. (2022a), in which the prompts are the outputs of a meta network with the visual features of each image as the inputs. In this paper, we want to ask the question, *why would the prompt learning methods or the improved conditional prompt learning methods have strong generalization ability?*

Before answering this question, we would like to take one step back and to think where does the generalization ability of a machine learning model comes from. Here we summarize three sources of the generalization ability of machine learning models.

1. **Knowledge**. Human knowledge is general. As a high level abstraction of human knowledge, natural language also has strong generalization power. The pre-trained language model and prompt engineering are some examples.

2. **Inductive bias**. During model design, experts can add some biases into the model based on the prior knowledge about the tasks to deal with. An example is the translation invariance of convolutional networks.

3. **Diverse training data.** The most simple and widely used method to improve generalization is to use large scale diverse data. The CLIP model is an example.

Despite that the learnable prompt vectors are optimized in the same word embedding space as NLP, the learned vectors are not natural language. Thus the prompt learning method can not generalize well to new categories that are not seen during training Zhou et al. (2022a). Thus conditional prompt learning is proposed which has shown high generalization ability than prompt learning. The question is where does the generalization power come from? Does it come from the introduced inductive bias or from the diverse training data? The first case (inductive bias) is hard to verify. Thus we will examine if the generalization power of conditional prompt learning comes from the optimization process.

Our assumption is that, due to the changed architecture, the improved prompt learning method may be actually trying to find a better optimality-generalization trade-off. If the prompt vectors are learned on the source dataset, then they will have poor generalization power on the target dataset. As a trade-off, the better the prompt learning fit the source dataset (optimality), the weaker generalization power it would have.

To verify our assumption, we conduct experiments using the original prompt learning method (CoOp). To find a better trade-off between optimality and generalization, we use a very simple method. Specifically, we train the CoOp model for various training epochs and compare the performance with conditional prompt learning method (we use the CoCoOp method in this paper). We use the same setting as in Zhou et al. (2022a) for all the experiments. The results are summarized in Figure 2.

Figure 2 shows that by controlling the optimality-generalization trade-off, the original prompt learning method (CoOp) can achieve higher generalization ability than conditional prompt learning method (CoCoOp) for most of the case. It seems that the learned prompts (either learned with or without conditional) are not related with natural languages. They are just some parameters which make parameter-efficient fine-tuning possible. All we need is to find a better optimality-generalization trade-off. These results highlight the need for the rethinking of existing prompt learning, and more careful baseline evaluations metrics are needed in future research on prompt learning methods in vision-language models.

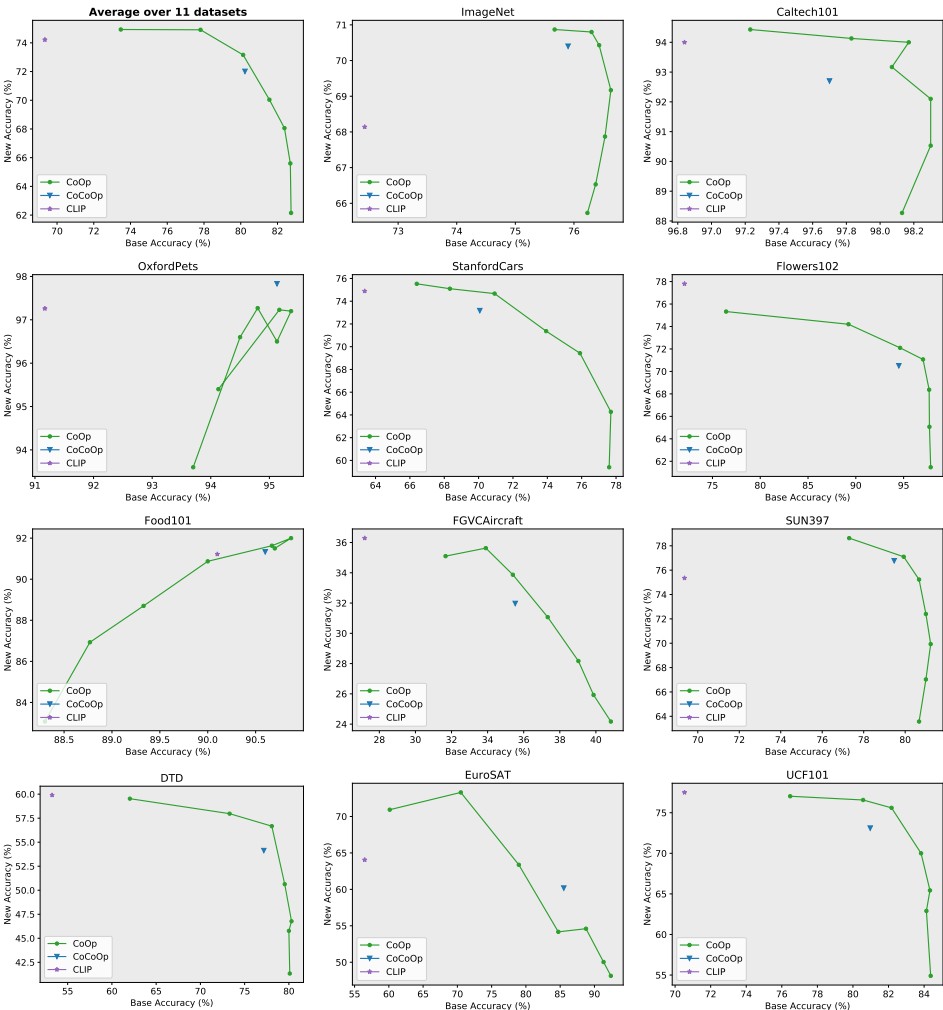

Figure 2: The trade-off between optimality and generalization.

## 7 CONCLUSION

This paper rethinks the existing prompt learning, making some surprising observations that contradict common beliefs about the prompt. First, we find that random prompts without fine-grained design or learning can also perform well in zero-shot recognition. Second, directly fine-tuning the linear classifier exhibits better performance than prompt learning. Moreover, we reveal that prompt learning is just a special case of parameter-efficient learning, and is a trade-off between optimality and generalization. Our results on 11 datasets highlight the rethinking in this paper can further boost the deployment of pre-trained vision-language models in downstream tasks.

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
