# OpenReview forum: "Rethinking the Value of Prompt Learning for Vision-Language Models"
_ICLR.cc/2023/Conference — Submitted to ICLR 2023_

### Official Review · Reviewer_2dEp · 2022-10-24

**Confidence:** 4
**Correctness:** 4
**Technical Novelty And Significance:** 2
**Empirical Novelty And Significance:** 2
**Recommendation:** 3

**Clarity, Quality, Novelty And Reproducibility:**

The paper could benefit from some extra work on the writing and exposition side. Some of the methods are not formally and thoroughly presented, and some of the results (tables or plots) do not contain complete captions and lack key contextual information.


**Strength And Weaknesses:**

Overall, the paper seems to be a set of interesting experiments, but with no clear takeaway. While I definitely think there's value in these experiments, I don't think the work is at the publication maturity level required by ICLR at the moment.

Questions.

- I don't think Linear Probe CLIP is formally defined or explained anywhere. What's the exact method here?
- In Table 4, what's the amount of few-shot examples per class used for finetuning?
- I think the classifier finetuning method isn't clearly explained, I was confused during a first reading. Maybe add a diagram or an example.
- I really struggled to understand Section 6. I think some major re-writing and experiment explanation is needed here.
- Figure 2 could benefit from a descriptive caption.

- Is the classifier finetuning method a new contribution? Or has this idea been tried elsewhere before? If it's a contribution, it should be made clear (and maybe even re-arrange the story and sections of the paper around it).

**Summary Of The Paper:**

This paper offers a set of experiments around prompting methods for contrastive vision-language models.

In particular, the following experiments are presented:

1) Handcrafted & random prompts.
A number of simple templates are used to do image classification on 11 datasets on top of a CLIP pre-trained model. In addition, two types of random prompts are included in the comparison. The conclusion is that handcrafted prompts do well --some better than others--, even those that oppose our language logic (i.e. "this is not a photo of") seem to do a fairly decent job. Random prompts are inferior, but still offer non-trivial (sometimes even strong) zero-shot performance.

2) Prompt learning versus Classifier Fine-tuning.
Moving away from zero-shot evaluation into the realm of supervised tasks, the paper compares prompt learning (adding learnable parameters for the embeddings of the prompt tokens + finetuning only those) and classifier fine-tuning. The latter, if I understood it correctly, rather than learning the original embeddings for the prompting tokens, directly learns the final representation for each possible class. These will be compared against the final embedding for the input image, and pick the closest class as usual in contrastive learning. Basically, the pre-trained language tower is ignored here. The experiment focuses on a few-shot regime (between 1-shot and 16-shot). It concludes that classifier fine-tuning is almost always superior to prompt learning, and also computationally cheaper / faster.

3) Robustness to Distribution Shifts.
Following up on 2), prompt learning and classifier fine-tuning are evaluated in terms of robustness. Table 4 shows the results of finetuning on ImageNet and then evaluating on other related (out-of-distribution) ImageNet datasets, for a variety of vision towers. While it is not reasonable to expect Zero-shot CLIP to perform as well as finetuned methods, we see that the classifier fine-tuning overall works best (more robust, in this case).

4) Optimality-Generalization Trade-off.
This one I didn't fully get. It seems to me one model is trained with CoOp for varying amounts of epochs, and results show that it dominates (in terms of original finetuning and out of distribution evaluation) a new technique (CoCoOp) that uses an auxiliary network.

**Summary Of The Review:**

The paper offers four interesting experiments around prompting (handcrafted, learned, finetuning "final class tokens", robustness, etc). While promising, I feel the paper still requires some work, including a more cohesive flow, and a bit more structured and unified takeaways.

---

> ### Author Response · Authors · 2022-11-26
> **Author Response to Reviewer 2dEp**
>
> We thank the reviewer for the valuable comments. Please find our response below:
>
> ## What's the exact method of Linear Probe CLIP?
> For linear probe CLIP, we follow the same training method used in CoOp and CLIP papers. The detailed method is as follows:
> 1. For K-shot classification, sample N = K * num_class images with target labels.
> 2. Extract the features of the N images
> 3. Train a logistic regression classifier on the extracted features and labels.
>
> ## In Table 4, what's the amount of few-shot examples per class used for finetuning?
> Thanks for pointing out this problem. We use 16 examples per class for finetuning, and evaluate on all validation/test set of the target dataset.
>
> ## The classifier finetuning method isn't clearly explained. Is the classifier finetuning method a new contribution?
> The classifier finetuning is the simplest transfer learning method of pre-trained models, i.e., we use the CLIP pre-trained vision backbone, add a fully-connected layer (the number of output neuron is the same to the class number), and then finetune the FC layer using CE loss.
>
> Classifier finetuning is not a new approach. It is the most widely used transfer learning from pre-trained dataset to downstream dataset for classification. Here we use classifier finetuning as a simple baseline, which can achieve better or the same accuracy than prompt learning.
>
> ## Figure 2 could benefit from a descriptive caption.
> Thanks for the suggestion. We revise the caption as follows:
>
> In prompt learning, the optimality (i.e., performance on base classes) and the generalization (i.e., performance on new classes) is a trade-off. The simple baseline of prompt leaning (CoOp) with various training epochs can provide better optimality-generalization trade-off than the sophisticated CoCoOp method. The results highlight the need for more careful evaluations on the generalization ability of prompt learning methods in vision-language models.

---

### Official Review · Reviewer_Tvm8 · 2022-10-26

**Confidence:** 5
**Correctness:** 2
**Technical Novelty And Significance:** 2
**Empirical Novelty And Significance:** 2
**Recommendation:** 3

**Clarity, Quality, Novelty And Reproducibility:**

Although the paper is well-written, the paper lacks clarity on what exactly it tries to study and mitigate. The experimental analysis is not extensive and lacks proper justification to them.

**Strength And Weaknesses:**

**Strengths**:

- The paper is well written and easy to follow.
- The paper has tried towards analyzing the effect of different compositional handcrafted prompts and studying the performance behavior on several downstream classification datasets.

**Weaknesses**:

- Although the paper states “Vision-Language Models” in its title, the experiments are only performed on CLIP models. It would be great to see similar findings for other vision-language models like DeCLIP, FILIP, CLOOB, CyCLIP, etc. Given that the paper is more of an analysis paper instead of a methodology paper, I would expect authors to verify their claims on other CLIP models like CLIP ViT variants beside CLIP ResNet. Does the size of models effect the conclusions presented in this paper? A more thorough comparison and anlysis should be included in the paper.
- It is interesting to observe that the four types of handcrafted templates used for zero-shot classification provide accuracies in a similar range, but this requires further extensive analysis to draw concrete conclusions. The {CLASS} token is very much important in classifying the images and is present in all the templates which provides the major information to the text encoder. The negative prompt has all the words the same as the revised prompt except “not”, obtaining accuracies in the similar range is expected as the text encoder is a language model and they lack compositionality which is well studied in NLP [a], and also the way CLIP has been pre-trained using the contrastive loss doesn’t guarantee compositionality. For the random prompts using random tokens and embeddings, the same argument of the presence of the {CLASS} token applies. Thus, a fair comparison between different prompts would be to remove the class token and then compare their performance on the downstream tasks. It might require some modifications at the output as done in "Learning to Decompose Visual Features with Latent Textual Prompts".
- The paper proposes classifier fine-tuning as a faster alternative to prompt learning and shows an average improvement of around 1.5%, but there is no parameter analysis (i.e., number of tunable parameters) provided for this experiment. It is known from the CoOp paper [b] that ensembling improves the performance, I assume that the authors fine-tuned the final projection layer of the text encoder which is of dimension 512x1024 for CLIP-ResNet-50, which has almost 64 times more parameters than learning 16 prompt tokens of 512 dimension. Therefore, it is very much essential to have a fair comparison with CoOp wrt the number of parameters each method uses. A fair comparison would be to use 64 learnable prompts and run CoOp for it. But, I believe from observing the results provided in CoOp [b], only 8 learnable prompts would be able to perform as good as the classifier fine-tuning. This argument applies to all the results in Figure-1 and Table-4.
- Additionally, the very motivation to learn continuous prompts for large vision and language models was to efficiently adapt the knowledge in the models to downstream tasks, which makes classifier fine-tuning contrasting to the motivation of prompt learning. I would like the authors to discuss on how classifier finetuning is a better alternative to prompt tuning for parameter efficient adaptation of large vision-language models for diverse downstream tasks.
- It would be good to add the average values across the datasets in table-2.
- No experimental comparison with other parameter efficient adaptation methods for CLIP, like CLIP-adapter [c], Tip-Adapter [d], UPL [e], PDL [f] etc. has been made in the paper. Authors should compare with these methods to verify the effectiveness of the proposed method over existing methods.
- Figure-2 is not clear. What are the epoch values corresponding to the points for CoOp?
- I feel that the section on optimality-generalization trade-off is very subjective and lacks proper experimental evaluation and theoretical support. In the beginning the authors ask the question - “why would the prompt learning methods or the improved conditional prompt learning methods have strong generalization ability?”, but get an answer - “Our assumption is that, due to the changed architecture, the improved prompt learning method maybe actually trying to find a better optimality-generalization trade-off.” I am confused on what it means. Additionally, I am not sure how the experiment of training CoOp for multiple epochs helps support the claims. Isn’t it known that training for a higher number of epochs makes networks susceptible to overfitting and is expected to lose generalizability? Also, why was only CoOp run for multiple epochs and not CoCoOp?

[a] Evaluating Compositionality of Sentence Representation Models: https://aclanthology.org/2020.repl4nlp-1.22.pdf

[b] Learning to Prompt for Vision-Language Models: https://arxiv.org/pdf/2109.01134.pdf

[c] CLIP-Adapter: Better Vision-Language Models with Feature Adapters: https://arxiv.org/pdf/2110.04544.pdf

[d] Tip-Adapter: Training-free CLIP-Adapter for Better Vision-Language Modeling: https://arxiv.org/pdf/2207.09519.pdf

[e] Unsupervised Prompt Learning for Vision-Language Models: https://arxiv.org/pdf/2204.03649.pdf

[f] Prompt Distribution Learning: https://arxiv.org/pdf/2205.03340.pdf





**Summary Of The Paper:**

The paper focuses on analyzing the prompt learning paradigm in CLIP. The authors discuss some observations on the use of various types of hand-crafted prompts including class names, basic prompts (e.g., "a photo of a {CLASS}"), negative prompts (e.g., “this is not a photo of a {CLASS}”), and random prompts on 11 benchmark datasets. The highlighted observations include no major drop in performance using negative prompts. Additionally, the paper analyzes classifier fine-tuning as a faster alternative to prompt learning and shows superior performance compared to prompt learning. Experiments on few-shot classification, robustness to distribution shifts, and generalization to unseen classes are included using the CLIP model as the backbone.

**Summary Of The Review:**

I think the identified problem is important but I’d like to rate the current submission as a clear rejection due to limited technical contributions and lack of convincing experiments. The paper needs significant changes including new experiments and possibly methodological improvements before being accepted to any major conference.

---

> ### Author Response · Authors · 2022-11-26
> **Author Response to Reviewer Tvm8**
>
> We thank the reviewer for the valuable comments. Please find our response below:
>
> ## Remove the class token and compare their performance on the downstream tasks.
> We thank the reviewer for the comments about the similar results of various handcrafted prompts. We agree with this analysis. We compare various handcrafted prompts for zero-shot setting, where the {CLASS} token is the only information to distinguish different classes, thus we didn't figure out how to remove {CLASS} token for zero-shot classification. It seems that the mentioned paper "Learning to Decompose Visual Features with Latent Textual Prompts" introduces learnable parameters and fine-tuning is needed.
>
> ## How classifier finetuning is a better alternative to prompt tuning for parameter efficient adaptation of large vision-language models for diverse downstream tasks.
>
> To answer this question, we need to recap this paper as a whole. In this paper, we rethink prompt learning from the following aspects.
>
> (1) In section 4, we point out the two common beliefs about prompt learning. We conduct experiments on various handcrafted prompts. First, random prompt can give good performance, which shows that prompt learning has a quite good start: even if nothing is learned, it can still achieve good performance. Thus, prompt learning is expected to be easily trained with few-shot examples. Second, handcrafted prompts generalize well, and the generalization ability for different prompts is consistent across different datasets. Thus, prompt learning, which learns in the prompt space, is expected to have powerful generalization ability. We check these beliefs in section 5 and section 6. Here we want to point out that the generalization ability only make sense for open-set classification/tasks.
>
> (2) For close-set classification, i.e., we train on class set A and test also on class set A, generalization ability doesn't make sense. We check if prompt learning is easily trained with few-shot examples than other methods. To this end, we design a simple baseline, i.e., the classifier finetuning. Results show that classifier finetuning achieves better accuracy than prompt learning with few-shot examples.
>
> (3) We check if prompt learning or the revised prompt learning method has better generalization ability for open-set classification. First, prompt learning does not generalize well for open-set classification. The reason is simple, i.e., the learned prompts overfit to the base classes. Previou works have shown these results so we didn't make further discussions. Second, does the improved prompt learning method (CoCoOp) solves the generalization problem of prompt learning? It seems so because CoCoOp outperforms CoOp on new classes. On the other hand, the improved performance on new classes is obtained with the cost of accuracy drop on base classes. Thus, we argue that CoCoOp doesn'st solve the generalization problem of prompt learning, but find a trade-off point between the optimality on base classes and the generalization on new classes. To this end, we again design a simple baseline method, the insufficiently trained CoOp (Insuf-CoOp), which is trained with various epochs (2, 5, 10, 25, 50, 100, 200). We plot the results in Figure 2. We can find that it is easily find a model that achieves similar result with CoCoOp on base classes but achieves much higher accuracy on new classes. Moreover, the results of Insuf-CoOp have justified our assumption, i.e., a model achieves good optimality on base classes usually results in bad generalization on new classes. Thus, to evaluate the generalization ability of prompt learning methods, more careful evaluations are needed. For example, we need to plot the pareto-frontier to see if the proposed method achieves better results than the simple baseline of Insuf-CoOp.
>
> ## Figure-2 is not clear. What are the epoch values corresponding to the points for CoOp?
> The accuracy curve from the start to the end corresponds to epoch (2, 5, 10, 25, 50, 100, 200). Note that, take the 50 epoch results for example, we train CoOp from scratch to the end for 50 epochs, instead of train the model for 200 epoch and use the accuracy at 50 epochs.

---

> > ### Comment · Reviewer_Tvm8 · 2022-12-05
> > **Final Response**
> >
> > I thank the authors for their response and effort on the new experiments. After carefully reading the authors response including other reviewers concerns, I am still not convinced with the key technical contributions of the paper. Even with the new experiments, I still feel the paper does not have enough contributions to be accepted for ICLR. The paper needs significant changes including new experiments and possibly methodological improvements before being accepted to any major conference. Based on all these points, I am keeping my initial rating on this paper.

---

### Official Review · Reviewer_pGd5 · 2022-11-03

**Confidence:** 5
**Correctness:** 2
**Technical Novelty And Significance:** 2
**Empirical Novelty And Significance:** 2
**Recommendation:** 3

**Clarity, Quality, Novelty And Reproducibility:**

I believe the paper is an analysis paper of existing prompt learning methods, but lacks clarity on the message it wants to convey, without proper justification of the claims, and comprehensive experimentation. No supplementary material and codes are provided for better reproducibility.

**Strength And Weaknesses:**

Strengths:

* The observations shown in the paper regarding the performance with various types of handcrafted prompts and their compositional variations is interesting.

Weaknesses/Questions/Suggestions:

* The experiments are only on the CLIP model, while the paper title says Vision and Language models.
* No parameter count included for classifier fine-tuning experiments. It is very much important to keep a track of the parameter count while proposing an alternative method for prompt tuning. An increase in parameters can increase the classification performance. How many parameters are tuned for classifier fine-tuning? By what number is it larger than CoOp? The parameter count needs to be the same for both the methods for an apples to apples comparison.
* What is the experimental setup for Figure-2? What are the epoch values?
* What is the justification for the experiment in Figure-2? Why is CoCoOp not run for multiple epochs?
* Experimental comparison with different prior works on prompt tuning is missing. Some examples include: Unsupervised Prompt Learning (UPL), Prompt Distribution Learning (ProDA), etc.


**Summary Of The Paper:**

The paper analyzes the behavior of CLIP towards various compositions of hand-crafted text prompts and puts out some observations. One of the key observations of the paper is the behavior of CLIP to negative prompts, in which the authors insert a “not” word to the prompt, e.g. “this is not a photo of a dog”. The paper shows that the zero-shot classification performance of CLIP is almost the same using negative prompts as using default prompts without the “not” word included. The paper also shows that the use of random tokens appended with the class name as prompts also give considerable performance. In the second part of the paper, the authors propose classifier fine-tuning in which they fine-tune the final layer of the text encoder instead of learning the prompts. The authors argue that this tackles the speed issues of prompt learning by not needing to backpropagate gradients through the whole of the CLIP model and gets better classification performance. Experiments include zero-shot classification using various hand crafted and random prompts, few-shot classification, robustness to distribution shifts and studies on optimality and generalization. Results are shown on 11 datasets.

**Summary Of The Review:**

All of my concerns are mentioned in the weaknesses section. I believe it is important to have clarity and proper experimental analysis which the paper lacks. I believe the paper needs to go through multiple iterations of revisions and rethinking on how to properly analyze and convey the findings for being fit for a publication at a venue like ICLR.

---

> ### Author Response · Authors · 2022-11-26
> **Author Response to Reviewer pGd5**
>
> We thank the reviewer for the valuable comments. Please find our response below:
>
> ## What is the experimental setup for Figure-2? What are the epoch values?
> We use the same hyper-parameters and experimental settings as used in the original paper of CoOp, but with different epochs. The CLIP backbone is ViT-B/16. The epoch values are [2 5 10 25 50 100 200].
>
> ## What is the justification for the experiment in Figure-2? Why is CoCoOp not run for multiple epochs?
> Through the prompt learning vs. classifier finetuning experiments, we have shown that for **close-set classification**, prompt learning does not fulfil the expectation of example efficient and more robust and generalizable.
>
> However, prompt learning can deal with open-set classification, which is not capable for classifier finetuning. Thus prompt learning has the potential to generalize well to new classes. Is this ture? Previous works have shown that prompt learning can not generalize well to new classes, the results of Figure 2 can also confirm this conclusion.  CoCoOp is proposed to solve this problem. It achieves much higher accuracy on new classes, at the cost of reduced performance on base classes. In this paper, we argue that 'single point' experiment is not enough, it can not conclude that CoCoOp is more generalizable than CoOp on new classes. Our assumption is that optimality (performance on base classes) and generalization (performance on new classes) is naturally a trade-off, i.e., a model with good base-class accuracy is expected to achieve bad new-class accuracy. To this end, again, we design a simple and trivil method, the insufficiently trained CoOp (Insuf-CoOp), which is trained with various epochs (2, 5, 10, 25, 50, 100, 200). We plot the results in Figure 2. We can find that it is easily find a model that achieves similar result with CoCoOp on base classes but achieves much higher accuracy on new classes. Moreover, the results of Insuf-CoOp have justified our assumption, i.e., a model achieves good optimality on base classes usually results in bad generalization on new classes. Thus to evaluate the generalization ability of prompt learning methods, more careful evaluations are needed. For example, we need to plot the pareto-frontier (instead of a 'single point') to see if the proposed method achieves better results than the simple baseline of Insuf-CoOp.
>
> ## Experimental comparison with different prior works on prompt tuning is missing.
> Thanks for this suggestion. We compare classifier fine-tuning with current parameter-efficient adaptation methods including UPL Huang et al. (2022), CoOp Zhou et al. (2022b), Tip-Adapter Zhang et al. (2021) and ProDA Lu et al. (2022). The results are summarized in Table 7. The number of images per class is 16. CLIP pre-trained model with ResNet-50 backbone is used for evaluation. The results show that classifier fine-tuning achieves the best results in 5 of the 11 downstream datasets.
>
> Method | ImageNet | Caltech101 | Pets | Cars | Flowers102 | Food101 | FGVC | SUN397 | DTD |	EuroSAT | UCF101 | Average |
> | ------ | -------- | ---------  | ---------- | ------------ | ---------- | ------- | ------------ | ------ | --- | ------- | ------ | ------- |
> | CLIP          | 58.2|  86.3|  85.8|  55.6|  66.1|  77.3|  17.3|  58.5|  42.3|  37.6|  61.5|  58.8|
> | UPL           | 60.5|  89.9|  88.3|  62.1|  68.9|  77.6|  17.3|  64.0|  46.6|  54.8|  67.2|  63.4|
> | UPL*          | 61.1|  91.4|  89.5|  71.0|  76.7|  77.9|  21.8|  66.4|  55.1|  71.0|  70.2|  68.4|
> | CoOp          | 63.0|  92.2|  89.2|  73.6|  94.6|  79.2|  30.9|  69.9|  63.8|  83.4|  76.2|  74.2|
> | Tip-Adapter   | 62.0|  90.2|  88.1|  66.8|  89.9|  77.8|  29.8|  66.9|  60.9|  70.5|  70.6|  70.3|
> | Tip-Adapter-F | **65.5**|  92.9|  89.7|  75.7|  94.8|  79.4|  35.6|  **71.5**|  66.6|  **84.5**|  **78.0**|  75.8|
> | ProDA         | 65.3|  91.3|  **90.0**|  75.5|  95.5|  82.4|  36.6|  -   |  **70.1**|  84.3|  -   |  -   |
> | Classifier FT | 64.7|  **93.4**|  89.2|  **76.5**|  **96.3**|  **79.2**|  **36.9**|  71.4|  66.8|  84.2|  77.9|  **76.0**|

---

### Official Review · Reviewer_4WBY · 2022-11-03

**Confidence:** 3
**Correctness:** 3
**Technical Novelty And Significance:** 2
**Empirical Novelty And Significance:** 3
**Recommendation:** 5

**Clarity, Quality, Novelty And Reproducibility:**

Weakness,

1. The results are sufficent and the conclusion are well established but the reasons behind the result need to be more explored. For example, why classifier fintuning is much better than learning prompt?

2. The novelty is limited but as a rethinking paper, it is fine.

3. Can other models except clip still support findings?


**Strength And Weaknesses:**

Strength
1. The experiments are sufficient and exhaustive.
2. The results provide the valuable hints that what's the better way to deploy the pretrained vision-language model.
3. The paper inspire people to think about more effective prompt design.


**Summary Of The Paper:**

In this work, prompt learning is reexamined, and several unexpected findings that defy accepted notions of the prompt are presented. First , random prompts without learning or fine-grained design may likewise function effectively in zero-shot recognition. Second, direct linear classifier fine-tuning performs more effectively than prompt learning. Furthermore, prompt learning is essentially a subset of parameter-efficient learning and represents a trade-off between generalization and optimality.  Findings across 11 datasets show that the approach presented in this research can significantly influnce the use of trained vision-language models in subsequent challenges.

**Summary Of The Review:**

Overall, this paper provides some valuable hints about prompts but the novelty is limited. Therefore, I give my initial rating as borderline

---

> ### Author Response · Authors · 2022-11-26
> **Author Response to Reviewer 4WBY**
>
> We thank the reviewer for the valuable comments. Please find our response below:
>
> ## The reasons behind the results need to be more explored.
>
> Thanks for the suggestion. We would like to try to explain the results from the following three aspects.
>
> (1) Results of various handcrafted prompts. As pointed out by reviewer Tvm8, the {CLASS} token is very important in classifying the images and is present in all the templates which provides the major information to the text encoder. The consistency of different handcrafted prompts on various datasets is because of the generalization nature of natural language.
>
> (2) Classifier finetuning outperforms prompt learning. In my opinion, classifier finetuning should be a better choice than prompt tuning for close-set classification, because classifier finetuning can directly adjust the target gradients of the text features (i.e., classifier finetuning directly adjust the text features towards a target value), while in prompt learning, it only can indirectly adjust the prompt vector to make a change on the text features towards a target value. More formally, given an image $x$ which belongs to the i-th class. The target of classifier finetuning is $maximize(cos(f(x), w_i)$ where $w_i$ are learnable parameters. While for prompt learning, the target is  $maximize(cos(f(x), g(t_i)))$ where $t_i$ are learnable parameters. Actually $w_i=g(t_i)$, $t_i$ can be viewed as the latent variables of $w_i$. If we can directly optimize $w_i$, there is no reason to optimize the latent variable of $t_i$.
>
> (3) Optimality on base classes and generalization on new classes is a trade-off. CLIP pretrained model generalize well because it has seen large scale of image-text pairs. Prompt learning on base classes is sure to improve accuracy on base classes, however, it will overfit to base classes. Thus, prompt learning on base classes will damage the generalization ability on new classes.

---

### Author Response · Authors · 2022-11-26
**Response to the common issues shared by reviewers (part 1)**

## The experiments are only on the CLIP model.
We have added more results. First, results of CLIP models with other backbones (ResNet-101, ViT-B/32, ViT-B/16) are given. Second, we also conduct experiments using DeCLIP Li et al. (2021) and CyCLIP Goel et al. (2022).

We provide additional results of classifier fine-tuning and prompt learning on CLIP with various backbones, i.e., ResNet-50, ResNet-101, ViT-B/32 and ViT-B/16. We use the 11 downstream datasets for evaluation. The number of images per class is 16. The results are summarized in the following table. The results over different backbones are consistent. On 10 of the 11 downstream tasks, classifier fine-tuning outperforms prompt learning, except on the OxfordPets dataset.

| Backbone | Method | ImageNet | Caltech101 | Pets | Cars | Flowers102 | Food101 | FGVC | SUN397 | DTD |	EuroSAT | UCF101 | Average |
| --------- | ------ | -------- | ---------  | ---------- | ------------ | ---------- | ------- | ------------ | ------ | --- | ------- | ------ | ------- |
| CLIP (ResNet-50) | Zero-Shot       | 58.2 | 86.3 | 85.8 | 55.6 | 66.1 | 77.3 | 17.3 | 58.5 | 42.3 | 37.6 | 61.5 | 58.8|
| CLIP (ResNet-50) | Prompt Learning | 63.0 | 92.2 | **89.2** | 73.6 | 94.6 | 79.2 | 30.9 | 69.9 | 63.8 | 83.4 | 76.2 | 74.2|
| CLIP (ResNet-50) | Classifier FT   | **64.7** | **93.4** | **89.2** | **76.5** | **96.3** | **79.2** | **36.9** | **71.4** | **66.8** | **84.2** | **77.9** | **76.0**|
| CLIP (ResNet-101) | Zero-Shot       | 61.6 | 89.8 | 86.8 | 66.2 | 64.0 | 80.5 | 18.4 | 59.0 | 38.6 | 32.6 | 61.0 | 59.9|
| CLIP (ResNet-101) | Prompt Learning | 66.5 | 94.1 | **90.6** | 79.9 | 95.2 | 82.2 | 34.5 | 71.4 | 66.0 | 83.5 | 78.5 | 76.6|
| CLIP (ResNet-101) | Classifier FT   | **67.5** | **94.7** | 90.0 | **81.9** | **96.7** | **82.5** | **39.8** | **72.9** | **68.6** | **84.9** | **81.4** | **78.3**|
| CLIP (ViT-B/32) | Zero-Shot       | 62.1 | 90.8 | 87.5 | 60.6 | 66.9 | 80.5 | 19.2 | 61.9 | 43.9 | 45.2 | 62.0 | 61.9|
| CLIP (ViT-B/32) | Prompt Learning | 66.8 | 95.2 | **91.1** | 75.9 | 95.1 | 82.0 | 32.2 | 73.1 | 66.1 | 82.4 | 78.9 | 76.3|
| CLIP (ViT-B/32) | Classifier FT   | **68.1** | **95.6** | 89.5 | **78.8** | **96.7** | **81.5** | **39.8** | **74.2** | **69.1** | **85.6** | **81.1** | **78.2**|
| CLIP (ViT-B/16) | Zero-Shot       | 66.7 | 92.9 | 89.2 | 65.3 | 71.3 | 86.1 | 24.7 | 62.5 | 44.4 | 47.6 | 66.8 | 65.2|
| CLIP (ViT-B/16) | Prompt Learning | 72.0 | 95.6 | **93.6** | 82.7 | 97.0 | 87.1 | 43.6 | 75.2 | 69.4 | 83.8 | 82.7 | 80.2|
| CLIP (ViT-B/16) | Classifier FT   | **72.9** | **95.9** | 92.8 | **84.2** | **98.1** | **87.3** | **47.9** | **76.6** | **72.5** | **87.9** | **84.6** | **81.9**|

We provide results with different vision-language pre-trained models. Specifically, we use three kind
of vision-language pre-trained models which are publicly available for evaluation, i.e., the CLIP,
DeCLIP Li et al. (2021) and CyCLIP Goel et al. (2022). The results are summarized in the following table. The
number of images per class is 16. The CyCLIP does not provide ViT-B/32 pre-trained model, thus
we use CyCLIP with ResNet-50 as backbone. Note that the results between different pre-training
methods are not comparable because different pre-training data is used. From the results, we can see
that classifier fine-tuning outperforms prompt learning for all pre-trained models and all downstream
tasks.

| Pre-train | Method | ImageNet | Caltech101 | Pets | Cars | Flowers102 | Food101 | FGVC | SUN397 | DTD |	EuroSAT | UCF101 | Average |
| --------- | ------ | -------- | ---------  | ---------- | ------------ | ---------- | ------- | ------------ | ------ | --- | ------- | ------ | ------- |
| CLIP (ViT-B/32) | Zero-Shot       | 62.1 | 90.8 | 87.5 | 60.6 | 66.9 | 80.5 | 19.2 | 61.9 | 43.9 | 45.2 | 62.0 | 61.9 |
| CLIP (ViT-B/32) | Prompt Learning | 66.8 | 95.2 | **91.1** | 75.9 | 95.1 | 82.0 | 32.2 | 73.1 | 66.1 | 82.4 | 78.9 | 76.3 |
| CLIP (ViT-B/32) | Classifier FT   | **68.1** | **95.6** | 89.5 | **78.8** | **96.7** | **81.5** | **39.8** | **74.2** | **69.1** | **85.6** | **81.1** | **78.2** |
| DeCLIP (ViT-B/32) | Zero-Shot       | 64.7 | 93.4 | 83.8 | 49.5 | 84.0 | 71.4 | 8.7  | 62.7 | 41.3 | 34.0 | 58.4 | 59.3|
| DeCLIP (ViT-B/32) | Prompt Learning | 66.9 | 95.9 | 87.6 | 66.6 | 98.6 | 73.8 | 25.1 | 70.0 | 67.1 | 83.3 | 74.6 | 73.6|
| DeCLIP (ViT-B/32) | Classifier FT   | **68.2** | **96.6** | **88.8** | **73.8** | **98.8** | **75.8** | **31.8** | **72.2** | **71.5** | **86.4** | **77.5** | **76.5**|
| CyCLIP (ResNet-50) | Zero-Shot       | 21.1 | 60.8 | 12.9 | 1.3 | 10.0 | 12.2 | 1.0  | 32.0 | 14.5 | 17.0 | 21.7 | 58.6|
| CyCLIP (ResNet-50) | Prompt Learning | 21.9 | 75.4 | 34.1 | 6.8 | 51.6 | 23.5 | 7.7  | 40.0 | 48.8 | 73.7 | 45.4 | 39.0|
| CyCLIP (ResNet-50) | Classifier FT   | **25.5** | **80.8** | **45.6** | **9.8** | **70.6** | **31.8** | **11.3** | **47.7** | **54.8** | **77.8** | **52.7** | **46.2**|

---

### Author Response · Authors · 2022-11-26
**Response to the common issues shared by reviewers (part 2)**

## There is no parameter analysis.

Thanks for pointing out this problem. We would like to answer this question from the following aspects.
1. Honest speaking, the extra parameter size is not the key in our rethinking paper. Our target is not to introduce an alternative method for prompt tuning. The information we mainly want to convey through the prompt learning vs. classifier fine-tuning experiment is that **prompt learning can not fulfil our common expectation**. (Here, our common beliefs are twofold, first, prompt learning converge faster and requires fewer training examples to train; second, prompt learning can provide strong robustness and generalization ability.) To support this view, we use the most simple and trivil method, i.e., adding a FC layer on top of the vision extractor, and finetine the FC layer (classifier finetuning). The results show that the trivil classifier finetuning achieves higher performance and robustness than prompt learning with few-shot training examples.
2.
   We provide the extra parameters introduced by prompt learning and classifier fine-tuning. We use CLIP model with ResNet-50 backbone for evaluation. The dimension of word embedding is 512, and the feature dimension for the vision/text branch is 1024.  For prompt learning with 16 tokens, the number of parameters are $16*512$ and $16 * 512 * num_{class}$for class-agnostic context (CAC) and class-specific context (CSC) versions of CoOp. For classifier finetuning, the parameter number is related to the class number, $1024 * num_{class}$. For downstream tasks, the class number is around 100.

    | num class | CLIP-RN50 | CoOp-CAC | CoOp-CSC | Classifier FT |
    | --------- | --------- | -------- | -------- | ------------- |
    |   100	    | 66.5 M | 8 K (0.012\%)| 0.8 M (1.2\%) | 0.1 M (0.15\%) |

    Thus the tunable parameters of classifier fine-tuning is about 12.5x larger than CoOp (CAC) and is about 8x smaller than CoOp (CSC). More detailed parameters of different parts on various downstream datasets are summarized in the following table.

    | Dataset | Num Class | TextEncoder | ImageEncoder | CoOp (CAC) | CoOp (CSC) | Classifier FT|
    | --- | --- | --- | --- | --- | --- | --- |
    | ImageNet   | 1000|  28.3 M|  38.2 M|  8 K|  8 M|  1 M|
    | Caltech101 | 100|  28.3 M|  38.2 M|  8 K|  800 K|  100 K|
    | OxfordPets | 37|  28.3 M|  38.2 M|  8 K|  296 K|  37 K|
    | StanfordCars | 196 | 28.3 M|  38.2 M|  8 K|  1.57 M|  196 K|
    | Flowers102 | 102|  28.3 M|  38.2 M|  8 K | 816 K | 102 K|
    | Food101    | 101|  28.3 M|  38.2 M|  8 K | 808 K | 101 K|
    | FGVCAircraft | 100|  28.3 M | 38.2 M | 8 K | 800 K | 100 K|
    | SUN397     | 397|  28.3 M|  38.2 M | 8 K | 3.18 M | 397 K|
    | DTD        | 47|  28.3 M | 38.2 M | 8 K | 376 K | 47 K|
    | EuroSAT    | 10 | 28.3 M | 38.2 M | 8 K | 80 K | 10 K|
    | UCF101     | 101 | 28.3 M | 38.2 M | 8 K | 808 K | 101 K|

    Moreover, we would like to further provide the following aspects:

    (1) Both CoOp and Classifier FT are parameter efficient. The small number of parameters introduced are negligible compared with the pre-trained models.

    (2) For classifier finetuning, there is no need to store the image encoder. Actually, for close-set classification considered in this section, it is not likely for CoOp to store the text encoder and learned prompts, because the classes are fixed, and it is more efficient to just store the text features (the dynamic weights mentioned in CLIP paper) for each class by discarding the text encoder. If this is the case, the CoOp and classifier finetuning will have the same extra storage (i.e., 1024 * num_class).

    (3) Classifier finetuning is more trainin-efficient, saving about 10x cost at training stage.
3. To make the comarision a bit more fair, we conduct experiments on CoOp with more parameters. Note that the extra parameters in classifier fine-tuning is fixed, thus we can not reduce the extra parameters of classifier finetuning method, thus we can only increase the extra parameters of CoOp. We also want to point out that the pre-trained CLIP allows at most 77 tokens, thus the prompt lenth must be smaller than 77 (the maximum prompt lenth depends on the maximum tokens in the class names of the downstream tasks.) We select the EuroSAT dataset for evaluation, which has 10 classes, for which we can control the prompt lenth to make the CoOp's parameter size larger than classifier FT. We also report ImageNet results with different number of prompts to see the impact of parameter size on the performance of CoOp. (Note that for ImageNet, the maximum prompt lenth allowed is 63. Thus we use 63 for the last row for ImageNet task.) The results are summarized in the following table.
    | Num Prompt | EuroSAT | ImageNet |
    | --- | --- | --- |
    | 4 | 82.2 | 63.2 |
    | 8 | 82.9 | 63.3 |
    | 16 | 83.2 | 63.0 |
    | 32 | 83.5 | 62.5 |
    | 48 | 83.5 | 62.5 |
    | 64 | 83.8 | 60.2 |

---

### Decision · Program_Chairs · 2023-01-20

**Decision:**

Reject

**Justification For Why Not Higher Score:**

All four reviewers recommend rejection. The AC agrees with this decision. In particular, some of the observations may not be so surprising as claimed by the authors (see the comments related to compositionality/negation by one of the reviewers). In addition, the paper would benefit from more analysis using the same number of parameters (extending the initial experiments provided in the author response)

**Justification For Why Not Lower Score:**

N/A

**Metareview: Summary, Strengths And Weaknesses:**

The paper presents a study on prompt learning, for example using negative or random prompts, and analyzes classifier fine-tuning as a faster and better alternative to prompt learning. While the analysis is interesting, and the paper is well written, all four reviewers raised major concerns regarding insufficient contributions, unfair comparisons with respect to number of parameters, and no clear takeaways. The author response was helpful, including for example new experiments on different vision-language models, but was not sufficient to eliminate the main reviewer's concerns.

**Summary Of Ac-Reviewer Meeting:**

N/A